# Antihypertensive Effects of *Gynura divaricata* (L.) DC in Rats with Renovascular Hypertension

**DOI:** 10.3390/nu12113321

**Published:** 2020-10-29

**Authors:** Mi Hyeon Hong, Xian Jun Jin, Jung Joo Yoon, Yun Jung Lee, Hyun Cheol Oh, Ho Sub Lee, Hye Yoom Kim, Dae Gill Kang

**Affiliations:** 1College of Oriental Medicine and Professional Graduate School of Oriental Medicine, Wonkwang University, Iksan, Jeonbuk 54538, Korea; mihyeon123@naver.com (M.H.H.); chinacross2013@hotmail.com (X.J.J.); mora16@naver.com (J.J.Y.); shrons@wku.ac.kr (Y.J.L.); host@wku.ac.kr (H.S.L.); 2Hanbang Cardio-renal Research Center & Professional Graduate School of Oriental Medicine, Wonkwang University, Iksan, Jeonbuk 54538, Korea; 3College of Pharmacy, Wonkwang University, Iksan 54538, Korea; hoh@wku.ac.kr

**Keywords:** renovascular hypertension, *Gynura divaricata* (L.) DC, renin-angiotensin-aldosterone system (RAAS), cardio-renal syndrome

## Abstract

*Gynura divaricata* (L.) DC (Compositae) (GD) could be found in various parts of Asia. It has been used as a traditional medicine to treat diabetes, high blood pressure, and other diseases, but its effects have not yet been scientifically confirmed. Therefore, we aimed at determining whether GD could affect renal function regulation, blood pressure, and the renin-angiotensin-aldosterone system (RAAS). Cardio-renal syndrome (CRS) is a disease caused by the interaction between the kidney and the cardiovascular system, where the acute or chronic dysfunction in one organ might induce acute or chronic dysfunction of the other. This study investigated whether GD could improve cardio-renal mutual in CRS type 4 model animals, two-kidney one-clip (2K1C) renal hypertensive rats. The experiments were performed on the following six experimental groups: control rats (CONT); 2K1C rats (negative control); OMT (Olmetec, 10 mg/kg/day)-treated 2K1C rats (positive control); and 2K1C rats treated with GD extracts in three different doses (50, 100, and 200 mg/kg/day) for three weeks by oral intake. Each group consisted of 10 rats. We measured the systolic blood pressure weekly using the tail-cuff method. Urine was also individually collected from the metabolic cage to investigate the effect of GD on the kidney function, monitoring urine volume, electrolyte, osmotic pressure, and creatinine levels from the collected urine. We observed that kidney weight and urine volume, which would both display typically increased values in non-treated 2K1C animals, significantly decreased following the GD treatment (^###^
*p* < 0.001 vs. 2K1C). Osmolality and electrolytes were measured in the urine to determine how renal excretory function, which is reduced in 2K1C rats, could be affected. We found that the GD treatment improved renal excretory function. Moreover, using periodic acid-Schiff staining, we confirmed that the GD treatment significantly reduced fibrosis, which is typically increased in 2K1C rats. Thus, we confirmed that the GD treatment improved kidney function in 2K1C rats. Meanwhile, we conducted blood pressure and vascular relaxation studies to determine if the GD treatment could improve cardiovascular function in 2K1C rats. The heart weight percentages of the left atrium and ventricle were significantly lower in GD-treated 2K1C rats than in non-treated 2K1C rats. These results showed that GD treatment reduced cardiac hypertrophy in 2K1C rats. Furthermore, the acetylcholine-, sodium nitroprusside-, and atrial natriuretic peptide-mediated reduction of vasodilation in 2K1C rat aortic rings was also ameliorated by GD treatment (GD 200 mg/kg/day; *p <* 0.01, *p <* 0.05, and *p <* 0.05 vs. 2K1C for vasodilation percentage in case of each compound). The mRNA expression in the 2K1C rat heart tissue showed that the GD treatment reduced brain-type natriuretic peptide and troponin T levels (*p <* 0.001 and *p <* 0.001 vs. 2K1C). In conclusion, this study showed that GD improved the cardiovascular and renal dysfunction observed in an innovative hypertension model, highlighting the potential of GD as a therapeutic agent for hypertension. These findings indicate that GD shows beneficial effects against high blood pressure by modulating the RAAS in the cardio-renal syndrome. Thus, it should be considered an effective traditional medicine in hypertension treatment.

## 1. Introduction

Cardiovascular dysfunction increases the incidence of heart failure, stroke, and hypertension. Fortunately, hypertension is known as the primary modifiable risk factor for cardiovascular disease [1,2]. Hypertension is associated with various diseases such as metabolic, renal, and endothelial dysfunctions [3]; it is responsible for sympathetic nerve activation, which, if excessive, causes various complications [4]. Endothelial cells are fundamental in blood circulation, vasoactive factor release, and vessel activation [5]. Nitric oxide is produced by nitric oxide synthase (NOS) in endothelial cells and is considered an important endothelium-derived relaxing factor. Reduced NO activity modifies endothelial status and also affects the cardiovascular system [6,7]. Reduced NO bioavailability could lead to cardiac and endothelial dysfunctions [8]. NO is also associated with kidney function by regulating glomerular ultrafiltration and medullary bloodstream [9,10]. Reactive oxygen species, produced by the impaired endothelin function, lead to a reduced NO availability [11]. Imbalanced NO production might impair mineralocorticoid functions or result in renal hypertension and cardiac dysfunction [12,13]. The two-kidney one-clip (2K1C) model used in this study is a classic renovascular hypertension model used in various experiments [14]. The renin-angiotensin-aldosterone system (RAAS) activation is crucial for the development and progression of hypertensive renal damage. The 2K1C model displays such hypertension due to the overactivation of the RAAS, leading to renal-heart damage [15]. The RAAS imbalance could lead to renovascular hypertension and several cardiovascular pathologies, as well as kidney dysfunction by increasing the angiotensin (Ang) II concentration and aldosterone level [16]. RAAS is an important regulator of renal blood flow, tubular sodium reabsorption, and Ang II activation. After the left kidney artery is clipped, the dysregulation of the RAAS triggers a sudden increase in Ang II concentration. Such an abnormal increase in the Ang II concentration results in the immoderate activation of the sympathetic nerves, functioning as a vasoconstrictor [17]. In addition, the 2K1C model is closely associated with NO production and increased oxidative stress, characterized by the presence of fibrosis, the deterioration of kidney structure and function [18,19], and subsequently, cardiac diseases [20]. Among the potential medicinal agents for hypertension, *Gynura divaricata* (L.) DC (Compositae) (GD), found in various parts of Asia, has long been used as a traditional medicine for hypertension and prescribed for diabetes treatment [21,22]. Several studies have shown that GD is associated with glucose metabolism and inhibits fat deposition in broilers [23,24]. In accretion, a similar plant, *Gynura procumbens*, has been reported to possess angiotensin-converting enzyme inhibitory activity [25]. However, no study has been conducted to evaluate the effects of GD extracts on a hypertensive rat model. Therefore, we aimed at investigating whether *G. divaricata* could ameliorate blood vessel dysfunction, kidney injury, or cardiac diseases in 2K1C rats and determining the *G. divaricate*-induced effector mechanisms in hypertension treatment.

## 2. Material and Methods

### 2.1. Plant Material and Gynura divaricata Extract Preparation

GD was purchased from the Misan Farm (Daegu, Korea) and boiled with 2 L of distilled water at 100 °C for 2 h. The GD extract was filtered using a filter paper and centrifuged at 900× *g* at 4 °C for 20 min, then the supernatant was incrassated in a rotary evaporator. We obtained 35.58 g of GD extract that we lyophilized using a freeze-drier and kept at −70 °C until further use.

### 2.2. Major Metabolite Isolation and Structural Analysis

The crude extract of GD (20.2 g) was suspended in distilled water (1000 mL) and successively partitioned with ethyl acetate and butanol (BuOH). The BuOH fraction (880 mg) was fractionated using reversed-phase C18 flash column chromatography (46 × 380 mm), eluting with a stepwise gradient of 10, 20, 30, 40, 50, 70, and 100 % (*v/v*) methanol (MeOH) in water (500 mL each) to produce seven consecutive fractions, GDB-1–7. The MWCB-2 subfraction was further purified by reversed-phase high-performance liquid chromatography using column YMC-Pack ODS-A (150 × 20 mm) eluting with a gradient of MeOH in water (0.1 % formic acid, 10 % to 40 % in 20 min) at a flow rate of 5 mL/min with ultraviolet (UV) detection at 254 nm to produce GDB-2-3 (2.6 mg, tR = 19.8 min) and GDB-2–4 (1.6 mg, tR = 21.2 min) (Appendix A).

### 2.3. 2K1C Hypertensive Rat Model

We used the previously reported 2K1C Goldblatt model to treat hypertension in this study [26]. Rats were anesthetized with intramuscular ketamine (25 mg/kg) and Rompun (5 mg/kg) treatments, We inserted a 0.2-mm-thick silver clip into the left renal artery to produce 2K1C hypertensive rats. All procedures were performed similar to those in case of the 2K1C rats in the control group, except for the silver clip application. On a weekly basis, we measured the body weight and tail-cuff systolic blood pressure (SBP), using tail-cuff plethysmography for the latter (MK2000; Muromachi Kikai, Tokyo, Japan), under optimal circumstances such as ambient temperature and quiet room. Measurements were performed at least determinations, and five-time averages were used for the experiment. The animals were divided into the following six experimental groups: control; 2K1C rats; 2K1C rats treated with olmetec (OMT, 10 mg/kg/day); and 2K1C rats treated with GD extracts at three different doses (50, 100, and 200 mg/kg/day) for 3 weeks by oral gavage. All groups were provided *ad libitum* with water and food (Cargill Agri Purina, Inc., Gunsan, Korea). Before being euthanized, six random rats in each group were individually housed in metabolic cages for 3 days (Tecniplast, Buguggiate, Italy). Their water and food intake were measured, and their urine samples were collected daily (24 h) in a plastic flask. At the end of the experiments, the investigated animals were anesthetized with 4% isofluorane in an inhaler, before being euthanized. All rats were sacrificed using the guillotine method, and trunk blood was collected in pre-chilled ethylenediaminetetraacetic acid (EDTA)-coated or heparinized tubes. All animal experiments were performed under the National Institute of Health Guidelines for the Care and Use of Laboratory Animals. The animal procedures were approved by the Institutional Ethics Committee for Animal Experimentation of Wonkwang University (approval number WKU18-5).

### 2.4. Aortic Tissue Immunohistochemistry

The thoracic aorta was separated in the rats, and the fat was carefully removed from the aorta for the prevention of endothelin cell damage. The aorta was cut into roughly 3-mm-wide rings and the experiment was implemented in O_2_ mix-containing Krebs solution (pH = 7.4). The relaxation response to acetylcholine (ACh) was measured in the aortic tissue precontracted using phenylephrine (PE, 1 μM). The isometric tension changes were recorded using a transducer (GrassFT 03, Grass Instrument Co., Quincy, MA, USA). Poly-L-lysine-coated slides (Fisher Scientific, Pittsburgh, PA, USA) were used for immunohistochemical staining. Paraffin sections were stained using the Histostain^®®^-SP kit (Invitrogen) following the streptavidin-biotin labeling method. We used the Image analysis software Image-Pro Plus for quantitative analysis, measuring an average of 10–20 randomly selected areas.

### 2.5. Aldosterone and Ang II Radioimmunoassay

We used a commercial aldosterone assay kit (Aldoctk-2, DiaSorin Inc., Stillwater, MN, USA) to measure aldosterone levels and analyzed them by radioimmunoassay (RIA). The Ang II measurement was performed using a commercial Ang II assay kit (Phoenix Pharmaceuticals, Inc., Burlingame, CA, USA) and it was also analyzed by RIA. Aldosterone and Ang II results were expressed as picograms of Ang II per milliliter (pg/mL).

### 2.6. Blood Analysis

Blood samples were obtained in a test tube containing EDTA and subsequently centrifuged at 3000 rpm for 20 min at 4 °C. Plasma samples obtained for albumin, blood urea nitrogen (BUN), and creatinine level measurements were frozen at −70 °C until further use and measured using a commercial kit (77184, Arkray, Japan).

### 2.7. RNA Isolation and Quantitative Real-Time Polymerase Chain Reaction (Q-PCR) Analysis

We used the RiboZol reagent (Amresco, Solon, OH, USA) for RNA extraction from the left atrium, right atrium, ventriculus sinister, and ventriculus dexter. We measured total RNA quantities by recording the absorbances of the samples at wavelengths of 260 and 280 nm using a UV spectrophotometer. Total RNA was reverse transcribed to cDNA using a High Capacity RNA-to-cDNA Kit (Applied Biosystems, Waltham, MA, USA). The Q-PCR analysis of the samples was performed using a Power SYBR Green PCR Master Mix (Applied Biosystems). The PCR primers used in this study are listed in Table 1. The gene expression levels were analyzed using a Step One Plus Real-Time PCR system (Applied Biosystems). The holding stage for gene amplification was started at 95 °C for 10 min to activate the AmpliTaq enzyme, followed by 40 cycles of amplification at 95 °C for 15 s (similar to denaturation), annealing, and extension at 60 °C for 60 s. The PCR product temperature was increased from 60 °C to 95 °C at a rate of 0.3 °C/s. The mRNA expression of each target gene was performed in triplicate and normalized to the endogenous glyceraldehyde 3-phosphate dehydrogenase (GAPDH).

### 2.8. Statistical Analysis

Bonferroni’s multiple comparisons test was used to compare significant differences by repeated measures of analysis of variance (ANOVA). Furthermore, we also applied the Student’s *t*-test for unpaired data. Statistical significance was defined as *p* < 0.05. The results are presented as the mean ± standard error (SE).

## 3. Results

### 3.1. High-Performance Liquid Chromatography Analysis of GD

GDB-2-3, GDB-2-4, GDB-4-1, and GDB-4-4 were identified as 2-propenoic acid, caffeic acid, dicaffeic acid, and kaempferol-3-O-rhamnoside, respectively, using ^1^H and ^13^C nuclear magnetic resonance (NMR) data analysis of the compounds and comparing our results with previously-reported data, as well as mass spectrum (MS) analysis in the case of GDB-4-1 and GDB-4-4 (Figure 1 and Appendix A).

### 3.2. GD Effect on Systolic Blood Pressure and Vascular Tension

We measured SBP in order to investigate how GD potentially affects it. The SBP of the control group and the 2K1C group were 105.2 ± 3.93 mmHg and 220.8 ± 4.95 mmHg, respectively. The SBP of the 2K1C group was higher than that of the control group but the SBP significantly decreased in the GD treatment group (Figure 2).

### 3.3. Effects of GD on Vascular Responses

The phenylephrine-pre-contracted aortic ring relaxation was 59.62 ± 7.22% in the 2K1C group. However, the vasodilation % changes after the acetylcholine treatment (Ach, 1 μM) were 76.46 ± 5.46, 86.26 ± 5.19, and 97.72 ± 8.08% in the case of the GD 50, 100, and 200 mg/kg/day treatments, respectively. The vascular response to ACh was better restored in the GD treatment group than in the 2K1C group (Figure 3A). The vasodilation percentage (%) response to the final concentrations of the ACh, sodium nitroprusside (SNP), and atrial natriuretic peptide (ANP) treatments are shown in Figure 3B–D. At the final SNP dose, the relaxation % of the 2K1C group (E_max_ 59.61 ± 2.22, logEC_50_ −7.79 ± 0.09) was lower than that of the control group (E_max_ 93.70 ± 2.83, logEC_50_ −8.81 ± 0.05). The relaxation % was higher in the 2K1C group treated with GD extracts at 100 and 200 mg/kg/day concentrations than in the 2K1C group. The maximal GD relaxation effect rates were 72.85 ± 4.49, 80.08 ± 4.17, and 76.4 ± 4.07% in the case of GD extract concentrations of 50, 100, and 200 mg/kg/day, respectively, and 88.9 ± 2.7% (OMT dose, 10 mg/kg/day) at 10 μM ANP concentration. In contrast, the maximal relaxation rate was 54.64 ± 7.28% at 10 μM ANP concentration. The related statistical comparison of ACh, SNP, and ANP is shown in Appendix A. The vascular relaxation of the aorta ring was decreased to a greater degree in the 2K1C group than in the control group. The aortic ring relaxation in the GD treatment group was more relevant than that in the 2K1C group (Figure 3B–D).

### 3.4. Effects of GD on Endothelial Nitric Oxide Synthase (eNOS) and Endothelin-1 (ET-1) Immunoreactivity in the Thoracic Aorta

In the thoracic aorta, endothelial nitric oxide synthase (eNOS) expression was measured using immunohistochemistry. The eNOS expression was lower in the 2K1C group compared to the control. However, the GD treatment significantly increased eNOS expression (Figure 4A). In Figure 4B, a digitalized eNOS image values show that the group treated with the GD extract at a concentration of 200 mg/kg/day exhibited higher eNOS expression (0.9657) than the 2K1C group (0.2586). Moreover, endothelin-1 (ET-1) immunoreactivity levels were higher in the 2K1C group than in the control. However, the 2K1C group treated with the GD extracts exhibited significantly higher ET-1 immunoreactivity levels than the control (Figure 4C). The digitalized ET-1 immunoreactivity was higher in the 2K1C group (2.7317) than in the control. However, the expression levels decreased in the group treated with the GD extract at a concentration of 200 mg/kg/day (1.0976) compared with the 2K1C group (Figure 4D).

### 3.5. Effects of GD on the Cardiac Function

In order to further clarify how GD might affect cardiac function, we measured the left ventricular weight, as well as brain natriuretic peptide (BNP) and troponin T (TnT) expression levels in GD-treated 2K1C rats. However, the ventricular weight/body weight ratios of the 2K1C group were significantly higher than those of the control group, and the GD groups showed significantly reduced ventricular weight/body weight ratios compared with those in the 2K1C group (Figure 5A,B). In order to investigate how GD might affect heart function, we measured the TnT and BNP expression levels, known as cardiac dysfunction biomarkers, in the left ventricle. The TnT and BNP mRNA levels were both higher in the 2K1C group than in the control group. The GD treatment suppressed the induction of these factors (Figure 6A,B).

### 3.6. Effects of GD on Renal Function

Aldosterone, angiotensin, BUN, albumin, and creatinine clearance (Ccr) are generally considered as renal function biomarkers, we thus measured them to evaluate how GD might affect renal function in 2K1C rats. We observed that BUN and aldosterone levels, as well as angiotensin concentration, were significantly lower in the GD-treated group than those in the 2K1C group. In addition, the albumin and Ccr levels in the GD-treated groups were also significantly more upregulated than those in the control group (Table 2). The urinary volume was significantly higher in the 2K1C group than in the control group and it significantly decreased in the GD group (Table 3). The urinary sodium, chloride, and potassium excretion levels significantly decreased in the 2K1C group. The electrolyte excretion level was better restored in the GD groups than in the 2K1C group (Table 3).

### 3.7. Effects of GD on Kidney Injury

The weight of the left clipped kidney was markedly lower in the 2K1C group than that in the control group (*p <* 0.001). In contrast, the weight of the right non-clipped kidney in the 2K1C group was significantly higher than that in the control group (*p <* 0.001). The administration of the GD extract (200 mg/kg/day) improved kidney weight (Appendix A). To determine how GD administration might affect tubular injury, we performed histological analysis. As shown by the PAS staining and density quantification, tubular damage could be observed in the 2K1C group. However, GD administration reduced fibrosis in the cortex (Figure 7), as well as in the inner (Figure 8) and outer medulla (Appendix A).

## 4. Discussion

Hypertension is a critical modifiable cardiovascular disease risk factor in our society [27]. Increased blood pressure is a critical point of heart disease, stroke, coronary heart disease, and renal disease, leading to millions of deaths annually [28]. RAAS activation is crucial for the development and progression of hypertensive renal damage. 2K1C causes hypertension by RAAS overactivation and hypertension-induced renal-heart damage [14]. Therefore, this study aimed at determining whether renal function regulation, blood pressure, and the RAAS were affected by GD in 2K1C renal hypertensive rats. According to various studies, the RAAS is the main cause of cardiovascular disease-related hypertension and body fluid imbalance-related renal disease [14,15]. Cardio-renal syndrome (CRS) refers to a disease caused by the interaction between the kidneys and the cardiovascular system, where the acute or chronic dysfunction in one organ might induce acute or chronic dysfunction of the other. In this study, we used the 2K1C model, a CRS type 4 model, to determine whether GD exhibited a cardio-renal mutual improvement effect in 2K1C.

Renal hypertrophy, which is fundamental to hypertensive renal damage, is one of the hallmarks of pathological changes. Renal hypertrophy is also an early biomarker of renal dysfunction, which indicates the presence of glomerular hyperperfusion, hyperfiltration, and glomerular hypertension [29]. We observed systemic blood pressure increase in the 2K1C rats, directly associated with glomerular perfusion and pressure, thus leading to proteinuria glomerulosclerosis, interstitial renal fibrosis, and renal hyperfiltration [30]. In this study, osmolality and electrolytes were measured in the urine to investigate renal excretory function following the GD treatment. The renal excretory function, which was reduced in 2K1C hypertensive rats, was improved by GD. Therefore, we suggest that GD treatment improves renal function in 2K1C.

With regard to vascular function, GD improved the acetylcholine-, sodium nitroprusside-, and atrial natriuretic peptide-induced impaired vascular dilation in the aorta in 2K1C rats. Endothelial dysfunction was first described in hypertension by the inhibition of NO activity. The vasorelaxation response to ACh was enhanced in the GD (50, 100, 200 mg/kg/day) treatment groups; NO release occurred via the activation of specific endothelial receptors [31,32]. NO and endothelin-1 (ET-1) are endothelium-derived mediators that play essential roles in vascular homeostasis [33]. Moreover, ET-1 is a crucial vasoactive substance associated with pathophysiological conditions such as hypertension, ischemic heart disease, and congestive HF [34]. In this study, immunohistochemistry analysis of the thoracic aorta showed decreased ET-1 levels following the GD treatment in 2K1C rats. These results suggest that the GD-induced restoration of the ET-1 receptor expression might be, at least in part, related to the impaired vascular reactivity improvement and lowering of the high blood pressure.

Ang II is a biological substance of RAAS and plays a critical role in renal function, such as renal disease, renal injury, and fibrosis [35]. 2K1C is well-known to increase blood pressure and activate the RAAS [36]. Our results showed that it was decreased following the GD treatment. 2K1C also increases aldosterone and Ang II levels. In this study, the plasma levels of these markers decreased following GD treatment. These results suggest that GD improves renal dysfunction by inhibiting the activity of the RAAS system.

The heart and the kidneys are closely associated with each other; therefore, their primary disorders extend to secondary dysfunction or injury [37]. Primary chronic kidney disease (CKD), a CRS type 4 condition, reduces heart function, resulting in ventricular hypertrophy and cardiovascular dysfunction [38,39]. CRS type 4 biomarkers, such as BNP and TnT, are used to diagnose cardiovascular diseases in patients with CKD [40,41,42]. Moreover, increased BNP and TnT levels are clinically used as cardiovascular mortality biomarkers [43,44]. In this study, the BNP and TnT mRNA expression levels in the heart tissues of 2K1C rats showed that their levels were reduced by the GD treatment. This study showed that GD improved cardiovascular and renal dysfunction observed in the presented innovative hypertension model, highlighting the potential of GD as a therapeutic agent for hypertension. However, it is necessary to acknowledge and resolve the two major limitations of the present study. First, we only studied the effect of GD, i.e., amelioration of the hypertension in the 2K1C mouse model, and did not study the associated underlying mechanisms. Second, in this study, we could only speculate that GD exhibits a protective effect signal on the RAAS in 2K1C mice. Therefore, more experiments should be performed in future studies to confirm the effect of GD on improving cardio-renal dysfunction.

## 5. Conclusions

This study showed that GD improved cardiovascular and renal dysfunction in an innovative hypertension model and provided important results that support the therapeutic utility of GD in treating hypertension. These findings indicate that GD exerts beneficial effects against high blood pressure by modulating the RAAS of the cardio-renal syndrome. Thus, GD could be considered an effective traditional medicine for hypertension treatment.

## Figures and Tables

**Figure 1 nutrients-12-03321-f001:**
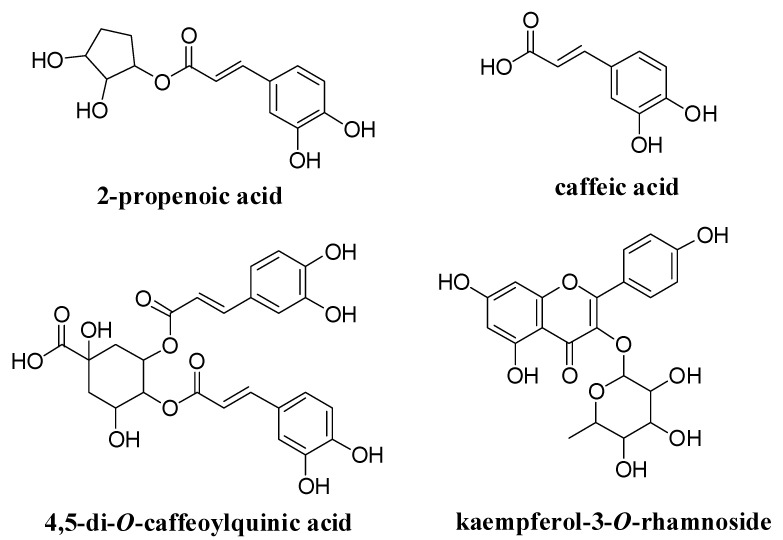
Structural identification of compounds isolated from *Gynura divaricata* (L.) DC (GD) using HPLC.

**Figure 2 nutrients-12-03321-f002:**
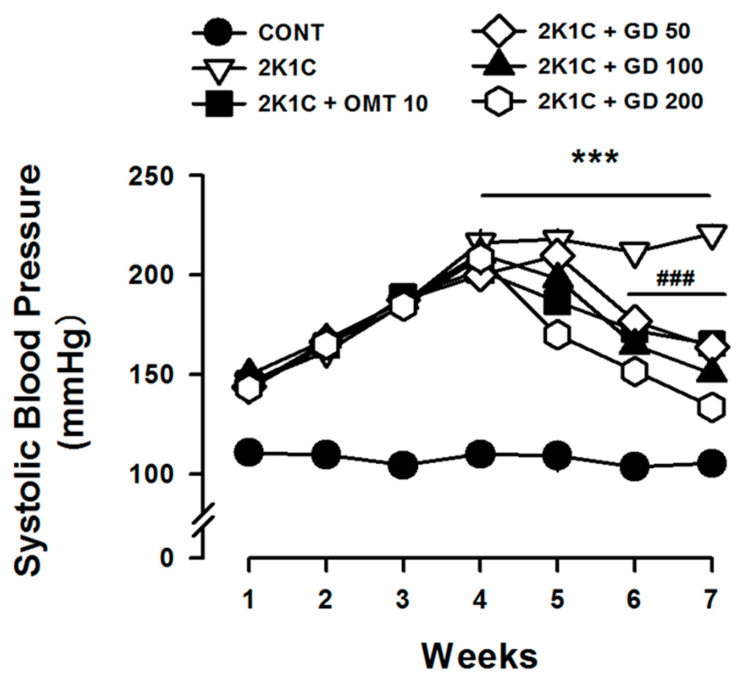
Effect of *Gynura divaricata* (L.) DC on systolic blood pressure (mmHg). The GD treatments at different concentrations (50, 100, 200 mg/kg/day) decreased systolic blood pressure (SBP) in 2K1C hypertensive rats. The values are expressed as the mean ± standard error (*n* = 10 per group). *** *p <* 0.001, vs. CONT and ^###^
*p* < 0.001, vs. 2K1C. 2K1C, two-kidney one-clip.

**Figure 3 nutrients-12-03321-f003:**
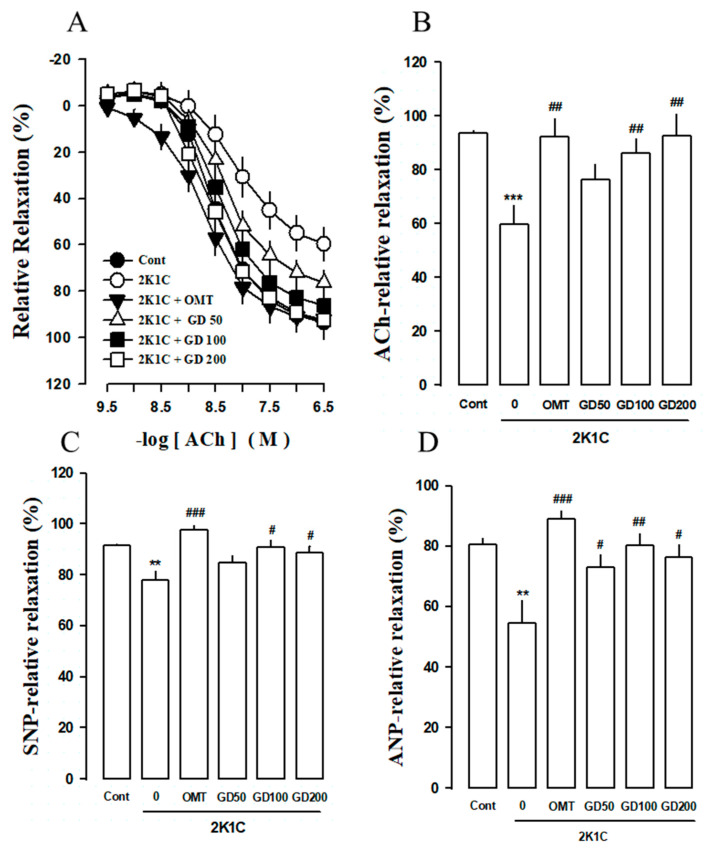
Vasorelaxation effect of *Gynura divaricata* (L.) DC (GD) following acetylcholine (ACh), sodium nitroprusside (SNP), and atrial natriuretic peptide (ANP) treatments. (**A**) Dose-response curves of ACh relaxation in the thoracic aorta of the 2K1C model; (**B**) The final ACh dose was changed in % change expression with each group; (**C**) Dose-response curves of SNP relaxation in the thoracic aorta of the 2K1C model; (**D**) Effect of GD on the ANP relaxation of the 2K1C model. Responses are expressed as the percentage of relaxations relative to the PE-induced pre-contractions. Each value shows the mean ± standard error (*n* = 10 per group). ** *p <* 0.01, *** *p <* 0.001, vs. CONT and ^#^
*p <* 0.05, ^##^
*p <* 0.01, ^###^
*p <* 0.0001 vs. 2K1C.

**Figure 4 nutrients-12-03321-f004:**
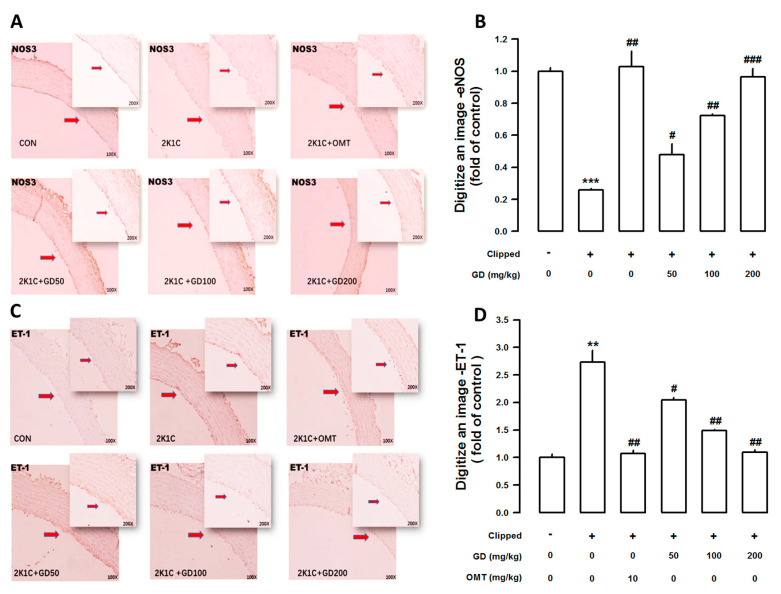
Effect of *Gynura divaricata* (L.) DC on endothelial nitric oxide synthase (eNOS) and endothelin-1 (ET-1) expressions in the blood vessels. (**A**) Immunohistochemical expression of eNOS in the aorta (magnification × 200); (**C**) Immunohistochemical expression of ET-1 in the aorta. Quantitative analysis of eNOS (**B**) and ET-1 (**D**) area. ** *p <* 0.01, *** *p <* 0.001, vs. CONT and ^#^
*p <* 0.05, ^##^
*p <* 0.01, ^###^
*p <* 0.0001 vs. 2K1C.

**Figure 5 nutrients-12-03321-f005:**
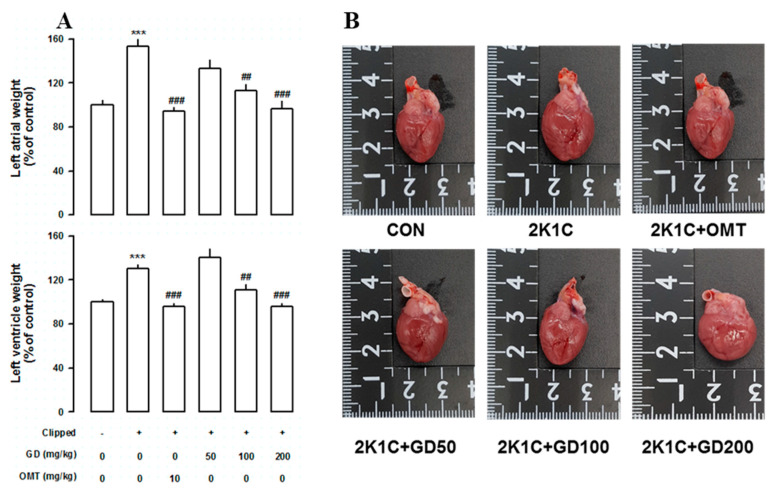
Effects of *Gynura divaricata* (L.) DC (GD) on heart weight and shape. (**A**) Effects of GD on the left atrial and left ventricular weight (expressed as % of control); (**B**) Photographs of the heart size in each group. Each value represents the mean ± standard error (*n* = 10 per group). *** *p <* 0.001 vs. CONT and ^##^
*p <* 0.01, ^###^
*p <* 0.001 vs. 2K1C.

**Figure 6 nutrients-12-03321-f006:**
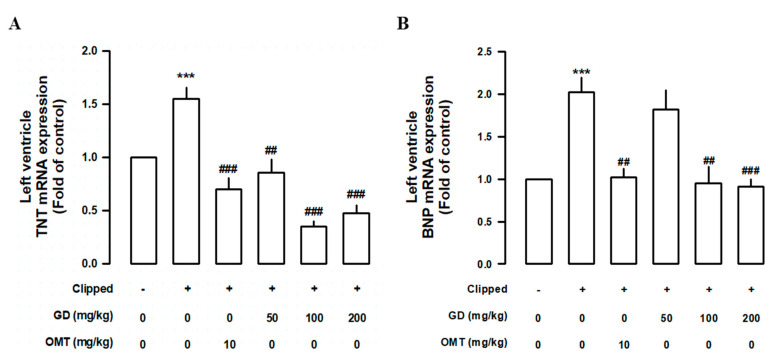
Effects of *Gynura divaricata* (L.) DC (GD) on the biomarkers on cardiac dysfunction. (**A**) Effects of GD on the expression of troponin T (TnT) mRNA. Real time quantitative polymerase chain reaction of TnT mRNA was performed from the left ventricle; (**B**) Effects of GD on the expression of brain natriuretic peptide mRNA. Each value shows the mean ± standard error (*n* = 10 per group). *** *p <* 0.001, vs. CONT and ^##^
*p <* 0.01 ^###^
*p <* 0.0001 vs. 2K1C.

**Figure 7 nutrients-12-03321-f007:**
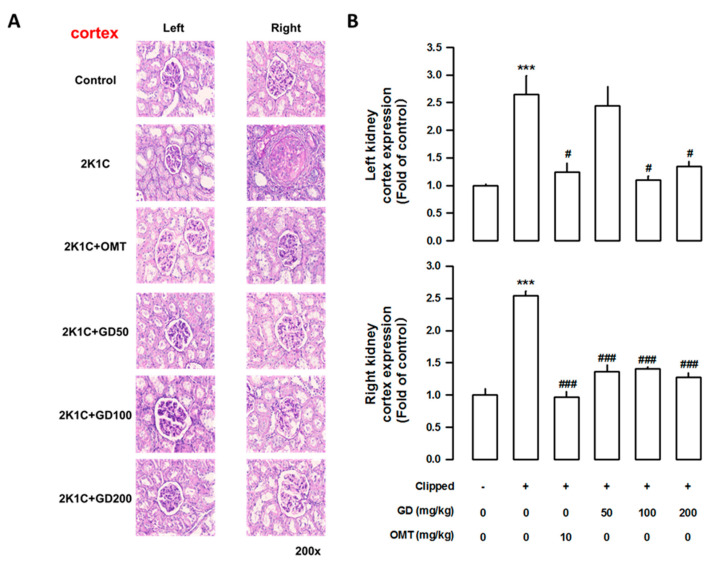
Effect of *Gynura divaricata* (L.) DC (GD) on fibrosis in the kidney tissue. (**A**) Periodic acid-Schiff (PAS) staining in the kidney cortex (magnification × 200); (**B**) Quantitative assessments representing the results of five independent experiments. The values are expressed as the mean ± standard error (*n* = 5 per group). *** *p <* 0.001, vs. CONT and ^#^
*p <* 0.05, ^###^
*p <* 0.0001 vs. 2K1C.

**Figure 8 nutrients-12-03321-f008:**
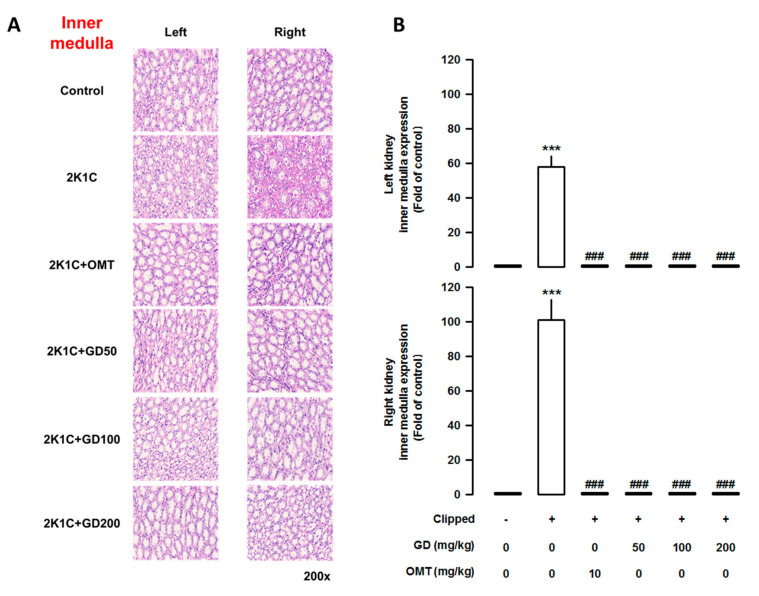
Effects of *Gynura divaricata* (L.) DC (GD) on kidney fibrosis in the outer medulla. (**A**) Periodic acid-Schiff staining of the kidney in the outer medulla. (magnification × 200); (**B**) Quantitative assessments representing the results of five independent experiments. The values are expressed as the mean ± standard error (*n* = 5 per group). *** *p <* 0.001, vs. CONT and ^###^
*p <* 0.0001 vs. 2K1C.

**Table 1 nutrients-12-03321-t001:** Primers used in the real-time polymerase chain reaction (Q-PCR).

Gene	Primer Nucleotide Sequence
*TNT*	Forward: 5′-CGT GGC TTG GGT TTG GTG T-3′ Reverse: 5′-AGA CTG GAG TGA AGA AGA GGA GGA C-3′
*BNP*	Forward: 5′-TGA TTC CTG CTC CTG CTT TTC-3′ Reverse: 5′-GTG GAT TGT TCT GGA GAC TG-3′
*GAPDH*	Forward: 5′-CAG TGC CAG CCT CGT CTC-3′ Reverse: 5′-AGG GGC CAT CCA CAG TCT-3′

**Table 2 nutrients-12-03321-t002:** Effects of the *Gynura divaricata* (L.) DC (GD) treatment on renal functional parameters.

	CONT	2K1C	OMT 10 (mg/kg/day)	GD 50 (mg/kg/day)	GD 100 (mg/kg/day)	GD 200 (mg/kg/day)
Aldosterone (pg/mL)	248.3 ± 54.5	1693 ± 496.6 ***	253.1 ± 46.3 ^###^	212.6 ± 69.5 ^###^	108.4 ± 13.6 ^###^	188 ± 20 ^###^
Angiotensin (pg/mL)	532.1 ± 36.8	1699.8 ± 144 ***	1168.6 ± 88.1 ^#^	1250.6 ± 119.5 ^#^	910 ± 109 ^##^	623 ± 112.6 ^###^
Albumin (g/dL)	3.1 ± 0.03	2.5 ± 0.06 ***	3.3 ± 0.05 ^###^	3.1 ± 0.05 ^###^	3.2 ± 0.06 ^###^	3.1 ± 0.12 ^##^
Bun (mg/dL)	15.7 ± 0.5	38.6 ± 7.9 **	16.5 ± 0.7 ^##^	18.8 ± 2.2 ^#^	18.2 ± 1.6 ^#^	18.2 ± 1.2 ^##^
Ccr (mL/min)	527.9 ± 143.2	35.2 ± 27.4 *	466.7 ± 100.1 ^##^	426.9 ± 63.7 ^#^	399.8 ± 100.6 ^##^	487.9 ± 92.2 ^##^

The values are expressed as the mean ± SE (*n* = 10 per group). * *p* < 0.01, ** *p* < 0.01, *** *p* < 0.01 vs. CONT. and ^#^
*p* < 0.05, ^##^
*p* < 0.01, and ^###^
*p* < 0.001 vs. 2K1C. Abbreviations: 2K1C, two-kidney one-clip, BUN, blood urea nitrogen; Ccr, creatinine clearance; OMT: olmetec.

**Table 3 nutrients-12-03321-t003:** Effects of the *Gynura divaricata* (L.) DC (GD) treatment on renal excretory function in 2K1C rats.

	CONT	2K1C	OMT 10 (mg/kg/day)	GD 50 (mg/kg/day)	GD 100 (mg/kg/day)	GD 200 (mg/kg/day)
Body weight (g)	458.6 ± 3.4	336.2 ± 18.1 ***	415 ± 7.3 ^##^	418.7 ± 6.3 ^##^	414.3 ± 3 ^##^	423.1 ± 7.8 ^##^
Water intake (m/day)	29.5 ± 1.8	61.5 ± 6.3 ***	38.5 ± 2.3 ^##^	38.8 ± 2.2 ^##^	29.5 ± 1.6 ^###^	30 ± 2.7 ^##^
Urine volume (g/23 h)	11.3 ± 0.8	47.3 ± 5.9 ***	19.8 ± 2.6 ^###^	17 ± 1.8 ^###^	15.5 ± 2 ^###^	15.6 ± 1.4 ^###^
Urine Cl^−^ (mmol/L)	274.5 ± 22.7	96 ± 6.1 ***	208.2 ± 36.5 ^#^	222 ± 15.4 ^###^	198.7 ± 14.4 ^###^	273 ± 31.1 ^###^
Urine Na^+^ (mmol/L)	165.7 ± 19.2	46.5 ± 6.1 ***	124.8 ± 19.3 ^##^	133.8 ± 11.9 ^###^	108 ± 10.6 ^##^	120.6 ± 23 ^##^
Urine K^+^ (mmol/L)	241.3 ± 18	78.3 ± 7 ***	182.5 ± 32.3 ^##^	208.9 ± 13.4 ^###^	186.7 ± 14.4 ^###^	201 ± 24.7 ^###^
Osmolality (mosm/kg)	1786 ± 111.1	570.8 ± 40.3 ***	1560.3 ± 241.4 ^###^	1540.2 ± 1.7.6 ^###^	1609 ± 151.3 ^###^	1597.8 ± 139.9 ^###^

The values are expressed as the mean ± SE. *** *p* < 0.01 vs. CONT. and ^#^
*p* < 0.05, ^##^
*p* < 0.01, and ^###^
*p* < 0.001 vs. 2K1C.

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
