# Peer review of "Antihypertensive Effects of Gynura divaricata (L.) DC in Rats with Renovascular Hypertension"

_nutrients, 2020, doi:10.3390/nu12113321_

Round 1
Reviewer 1 Report
I am really impressed by these outstanding results.
Apart the awful english and the fact that gynura divaricata has already shown its antihypertensive effects (1), I have few concerns:
- Why the authors do not speculate on the mechanisms of action of the herb extracts?
- Is there a direct vasodilator action underlining the effects of GD estracts? In this case the prevention/regression of the LVH here shown is quite in contrast with the available literature.
- Is there a possible calcium channel interference (as already reported in literature)?
- Is there a sort of possible inhibition of the RAAS ? Again, in this case the levels of Aldosterone and Renin do not warrant the hypothesis.
- Any control of the dietary salt in the groups of animals?
- The authors show a scarce knowledge of western author names, when they cite previous papers with the first name of the authors instead of family names…
- Incidentally, hypertension is not a main risk factor, instead it is the primary modifiable risk factor for cardiovascular risk.
- Incidentally, the renin-angiotensin-aldosterone system is not the RAS, but the RAAS.
If the authors can discuss convincingly (and I mean very concincingly) all these points I would be prepared to evaluate a revised version.
Finally the authors should include in the reference section the paper here cited, Ref. (1):
References
- In vitro studies of Gynura divaricata (L.) DC extracts as inhibitors of key enzymes relevant for type 2 diabetes and hypertension. Wu T, Zhou X, Deng Y, Jing Q, Li M, Yuan L. J Ethnopharmacol. 2011; 136: 305-308. doi: 10.1016/j.jep.2011.04.059.
Author Response
Comments and Suggestions for Authors
I am really impressed by these outstanding results.
Apart the awful english and the fact that gynura divaricata has already shown its antihypertensive effects (1), I have few concerns:
I really thank you very much for your invaluable suggestions and comments.
Comment 1: Why the authors do not speculate on the mechanisms of action of the herb extracts?
Response 1: We studied the 2K1C hypertension animal model, the cardio-renal syndrome type 4 model, in which the kidneys and the heart are affected together, focusing on whether it is improved by GD. As the reviewer suggested, we do not study the mechanisms of GD. We performed only the study to evaluate the GD effect on the hypertension model first of all. It would have been nice if we had conducted the GD is mechanism research, but unfortunately, we conducted the efficacy analysis first, and we think that the mechanism research should be further studies. Thus, we have mentioned in the Discussion section about the limitations of these studies, should be performed in further studies (page 14, section 4).
Comment 2: Is there a direct vasodilator action underlining the effects of GD estracts? In this case the prevention/regression of the LVH here shown is quite in contrast with the available literature.
Response 2: There is no article about detailed vasorelaxation effect of GD extracts. In this study, figure3 and figure4 showed that GD increased relaxation percentage of blood vessel and attenuated blood vessel function in 2K1C hypertensive rats. We discovered that GD has been shown to be effective in blood vessel function. As with other studies, we were able to identify the same findings that improved vascular relaxation by drug effects at 2K1C1,2). We will perform further study to evaluated mechanism of GD vasodilation.
1) S. Choi, W.S. Jung, N.S. Cho, K.H. Ryu, J.Y. Jun, B.C. Shin, J.H. Chung, C.H. Yeum. Mechanisms of phytoestrogen biochanin A-induced vasorelaxation in renovascular hypertensive rats. 2014; 33:181-186.
2) D.A. Guimaraes, E. Rizzi, C.S. Ceron, A.M. Oliveira, D.M. Oliveira, M.M. Castro, C.R. Tirapelli, R.F. Gerlach, J.E. Tanus‐Santos. Doxycycline Dose‐dependently Inhibits MMP‐2‐Mediated Vascular Changes in 2K1C Hypertension. Basic Clin Pharmacol Toxicol. 2011;108(5):318-25.
Comment 3: Is there a possible calcium channel interference (as already reported in literature)?
Response 3: Various studies have reported that the hypertension model is associated with calcium channels1,2). Therefore, there is a possibility of calcium channel interface in this study. Unfortunately, we haven't studied calcium channels, so the correlation between GD and calcium channels has yet to be confirmed. As the reviewer suggest, further research is likely to be necessary because there are clear limitations to research. Thus, we have mentioned in the Discussion section about the limitations of these studies (page 14, section 4).
1) A. Mori, S. Suzuki, K. Sakamoto, T. Nakahara, K. Ishii. Role of calcium-activated potassium channels in acetylcholine-induced vasodilation of rat retinal arterioles in vivo. Naunyn Schmiedebergs Arch Pharmacol. 2011;383:27-34.
2) GE Callera, A Yogi, R.C. Tostes, LV Rossoni, LM Bendhack. Ca2+-activated K+ channels underlying the impaired acetylcholine-induced vasodilation in 2K-1C hypertensive rats. J Pharmacol Exp Ther. 2004;309:1036-42.
Comment 2: Is there a sort of possible inhibition of the RAAS? Again, in this case the levels of Aldosterone and Renin do not warrant the hypothesis.
Response 2: According to various research reports, after clipping renal artery, plasma renin activity, aldosterone, and angiotensin level were increased1). As with other studies, our result showed tendency that angiotensin, aldosterone level increased in 2K1C renovascular hypertensive rats2).
Figure. Vascular clipping and unclipping resulted in plasma renin activity changes.
Figure. Angiotensin-dependent mechanisms activated by unilateral renal arterial stenosis.
After renal arterial stenosis, marked increases in renin formation by the clipped kidney lead to increases in circulating angiotensin II (ANG II), thus leading to various ANG II-mediated changes as shown in Figure. As explained in the text, ANG II content in the renin depleted non stenotic kidney also increases through mechanisms not dependent on renin2).
Figure. Plasma and kidney angiotensin II (ANG II) levels in 2-kidney, 1-clip (2K1C) Goldblatt hypertensive rats and ANG II-infused rats.
There is RAAS antagonist such as angiotensin converting enzyme inhibitor or angiotensin receptor blocker to cure hypertension and cardiovascular disease3). We used olmesartan, which was used to treat hypertension, heart failure. Our results suggested that GD has been shown to be similar with olmesartan effect in 2K1C renovascular disease. Thus, we supposed to GD could have the possibility as angiotensin antagonist in renovascular hypertension.
1) LQ Li, J Zhang, R Wang, JX Li, YQ Gu. Establishment and evaluation of a reversible two-kidney, one-clip renovascular hypertensive rat model. Exp Ther Med. 2017; 13:3291-3296.
2) L.G. Navar, L Zou, A.V. Thun, C.T. Wang, J.D. Imig, K.D. Mitchell. Unraveling the Mystery of Goldblatt Hypertension. News Physiol Sci. 1998; 13:170-176.
3) C.A. Romero, M. Orias, M.R. Weir. Novel RAAS agonists and antagonists: clinical applications and controversies. Nat Rev Endocrinol. 2015; 11:242-252.
Comment 3: Any control of the dietary salt in the groups of animals?
Response 3: We didn’t salt diet in each group. We did not control salt intake by diet control, and we conducted the study with the same general diet in all groups.
Comment 4: The authors show a scarce knowledge of western author names, when they cite previous papers with the first name of the authors instead of family names…
Response 4: We revised all the wrong author name.
Comment 5: Incidentally, hypertension is not a main risk factor, instead it is the primary modifiable risk factor for cardiovascular risk.
Response 5: We have revised it.
Comment 6: Incidentally, the renin-angiotensin-aldosterone system is not the RAS, but the RAAS.
Response 6: We have revised to RAAS.
We really thank Reviewer #1 very much indeed.
If the authors can discuss convincingly (and I mean very concincingly) all these points I would be prepared to evaluate a revised version.
Finally the authors should include in the reference section the paper here cited, Ref. (1):
Reference
1.In vitro studies of Gynura divaricata (L.) DC extracts as inhibitors of key enzymes relevant for type 2 diabetes and hypertension. Wu T, Zhou X, Deng Y, Jing Q, Li M, Yuan L. J Ethnopharmacol. 2011; 136: 305-308. doi: 10.1016/j.jep.2011.04.059.

Reviewer 2 Report
My general comments:
Methods: The treatments were administered via "oral intake" - by this do you mean via oral gavage? Or were the treatments in food or water? This must be clarified.
Rats were euthanased by guillotine (note that the word sacrificed should not be used): were the rats anaesthetised prior to guillotine (I sincerely hope so)? This must be stated in the methods.
There is no description of the isolated aortic rings setup or protocol. What were the tissues preconstricted with for example?
Fig. 2 - the treatments should be explained in the legend.
Fig. 3 - EC50 values should be obtained for each individual tissue's ACh curve and statistically compared between treatment groups; i.e. the sensitivity to ACh vasorelaxation in each treatment group. Fig. 3 B-D should be more correctly described as Emax responses to each agonist. It looks by eye that there would be no change in sensitivity to ACh, but only to the maximum response (Emax).
The Discussion needs to be extensively rewritten with improvements to the English grammar (this applies to the whole manuscript). It is difficult to appreciate the authors' ideas in the current version.
Author Response
Comments and Suggestions for Authors
I really thank you very much for your invaluable suggestions and comments.
My general comments:
Comment 1: Methods: The treatments were administered via "oral intake" - by this do you mean via oral gavage? Or were the treatments in food or water? This must be clarified.
Response 1: We administered the drug by oral gavage. we have revised the text in Material and Methods with the oral gavage (page 3, section 2.3).
Comment 2: Rats were euthanased by guillotine (note that the word sacrificed should not be used): were the rats anaesthetised prior to guillotine (I sincerely hope so)? This must be stated in the methods.
Response 2: In our study, animals that were anesthetized with 4 % isoflurane in inhalers before euthanasia was used. Thus, Method 2.3 was revised.
Comment 3: There is no description of the isolated aortic rings setup or protocol. What were the tissues preconstricted with for example?
Response 3: We added the detailed method about tension. The thoracic aorta was separated in rat and fat in aorta were removed carefully for prevention of endothelin cell damage. The aorta cut into rings of roughly 3 mm wide and the experiment was implemented in Krebs’ solution (pH 7.4) containing O2 mix. The relaxation of aortic tissue was measured response to ACh dose-independent in rings precontracted by PE (1 μM) (page4, section 2.4).
Comment 4: Fig. 2 - the treatments should be explained in the legend.
Response 4: We have revised to figure legend.
Figure 2. Effect of GD on systolic blood pressure (mmHg). GD treatments decreased SBP in 2K1C hypertensive rats. Vales are expressed as mean ± S.E. (n=10 per group). *** p<0.001, vs. CON; ###<0.001, vs. 2K1C.
Comment 5: Fig. 3 - EC50 values should be obtained for each individual tissue's ACh curve and statistically compared between treatment groups; i.e. the sensitivity to ACh vasorelaxation in each treatment group. Fig. 3 B-D should be more correctly described as Emax responses to each agonist. It looks by eye that there would be no change in sensitivity to ACh, but only to the maximum response (Emax).
Response 5: As the reviewer suggested, we made a statistical comparison of ACh, SNP, and ANP. The related information was mentioned in Results and entered as supplementary data (page 6, section 3.3).
Supplemental data 2. EC50 and Emax values for ACh, SNP, and ANP-induced relaxation of GD treatment in 2K1C rats
|
|
ACh |
SNP |
ANP |
|||
|
Emax (%) |
log EC50 (M) |
Emax (%) |
log EC50 (M) |
Emax (%) |
log EC50 (M) |
|
|
cont |
93.70±2.83 |
-8.81±0.05 |
91.34±0.54 |
-8.99±0.03 |
80.60±2.00 |
-8.44±0.28 |
|
2K1C |
59.61±2.22*** |
-7.79±0.09* |
77.72±3.77** |
-8.14±0.07* |
54.64±7.28** |
-8.00±0.31* |
|
2K1C+OMT |
92.22±3.79## |
-8.45±0.17# |
97.72±1.52### |
-8.88±0.02# |
88.91±2.71### |
-8.69±0.06# |
|
2K1C+GD50 |
76.45±3.46 |
-8.09±0.07 |
84.71±2.76 |
-8.66±0.05 |
72.85±4.49# |
-8.51±0.08# |
|
2K1C+GD100 |
86.28±3.12## |
-8.12±0.08 |
90.96±2.88# |
-8.84±0.03# |
80.09±4.17## |
-9.36±0.10# |
|
2K1C+GD200 |
92.72±4.08## |
-9.03±0.14# |
88.74±2.43# |
-8.94±0.03# |
76.41±4.07# |
-9.36±0.09# |
EC50 and Emax values of ACh, SNP, and ANP were calculated of the relaxation-response curves to GD treatment in 2K1C rats. Values are means ±SE of 6 experiments for EC50 and Emax. *p < 0.01, **p < 0.01 ***p < 0.01 vs. Cont.; # p < 0.05, ##p < 0.01, and ### p < 0.001 vs. 2K1C
Comment 6: The Discussion needs to be extensively rewritten with improvements to the English grammar (this applies to the whole manuscript). It is difficult to appreciate the authors' ideas in the current version.
Response 6: Thanks for reviewing the manuscript submitted. We tried to revise this manuscript grammar overall. Therefore, some fatal grammar errors have been corrected. If we need better proofreading, we can apply to a professional editing company institution anytime.
We really thank Reviewer #2 very much indeed.

Round 2
Reviewer 1 Report
English remains a major trouble in this paper. I suggest the authors to copy-edit the full text together with a native language speaker.
The Reference section is unacceptable: names of the authors in cited papers remain wrong. Is this a joke ?
Rows 349-352 are extremely confused and should be completely edited and re-written.
Finally I can't find the citation I suggested to add to references... see the first review.
Author Response
Comment 1. English remains a major trouble in this paper. I suggest the authors to copy-edit the full text together with a native language speaker.
Response 1. I really thank you very much for your invaluable suggestions and comments. We tried to revise this manuscript grammar overall. Therefore, some fatal grammar errors have been corrected. However, a given day was too short to be modified by specialized institution. If we need better proofreading, we can apply to a professional editing company institution anytime.
Comment 2. The Reference section is unacceptable: names of the authors in cited papers remain wrong. Is this a joke ?
Response 2. I'm so sorry. We made a big mistake not to revise the References completely. We have revised the References as suggested.
Comment 3. Rows 349-352 are extremely confused and should be completely edited and re-written.
Response 3. We have revised the text as suggested.
Comment 4. Finally I can't find the citation I suggested to add to references... see the first review.
Response 4. We have added to references as suggested.
Thanks for reviewing the manuscript submitted, and we really thank Reviewer #1 very much indeed.
Reviewer 2 Report
Thank you for making most of the changes recommended in my review.
Author Response
Thanks for reviewing the manuscript submitted, and we really thank Reviewer #2 very much indeed.